# An Outlook of Drilling Technologies and Innovations: Present Status and Future Trends

**Catalin Teodoriu** * and **Opeyemi Bello**

Mewbourne School of Petroleum and Geological Engineering, The University of Oklahoma, Norman, OK 73019, USA; Opeyemi.O.Bello-1@ou.edu
* Correspondence: cteodoriu@ou.edu

**Abstract:** The present article analyzes the technological advancement and innovations related to drilling operations. It covers the review of currently proven and emerging technologies that could mitigate the drilling operational deficiencies and instabilities that could hinder operational performance activities and the economic part of drilling development with great effort to minimize their environmental footprint. Drilling system design and operations are among the major aspects and cost-effective endeavors of the oil and gas industries, which are therefore technology dependent. They are also considered to be among the most expensive operations in the world, as they require huge expenses daily. Drilling success, depending on prevalent conditions, is a function of several general factors. These include the selection of the best technologies and tools, procedural optimization, concrete problem-solving, accurate prediction, and rapid decision-making. Consequently, any sorts of tools or advanced technologies that can improve the time-efficient operational and economic performance of drilling activities are essential and demanded. The paper provides a review of available technologies and developmental innovations based on both company-based and academic research-enabled drilling solutions over the past 5 years in the field of drilling systems and technological design. The paper further highlighted potential technologies that could be tapped in from other industries and could possibly be adopted by pushing the conventional boundaries of drilling operations.

**Keywords:** drilling; advanced technologies; emerging technologies; operational and economic performance; environmental footprint



## 1. Introduction

The world's need for energy demand and transformation towards harvesting hydrocarbons is accelerating. This is attributable to a combination of technological progress, development priorities, and environmental concerns. The drilling operation is one of the most sensitive divisions in the oil and gas industry. With geothermal energy as a stable energy resource, deep drilling at attractive costs is pushed every day to the limits. Advancements in the processes, techniques, technologies, and innovations associated with well drilling activities all have significant impacts on drilling operational efficiency, safety, and economy, which must be maintained within acceptable levels in unlocking hydrocarbon resources.

Subsurface resource development is subject to drilling wells in safe and cost-efficient manners. Since the discovery of the first oil well in Titusville by Edwin Drake and the progression of the petroleum business over the years, the cable tool and rotary drilling were the first techniques applied in the drilling phase. In recent years, drilling activities for searching hydrocarbons have shown considerable technological advances in order to obtain safe, environmentally friendly, and cost-effective well construction with increased performance. Some of the early advancements in drilling technology seeking innovative approaches towards better drilling efficiency were based on three basic factors: safety, minimum, and usable hole. These technologies include horizontal drilling, multilateral drilling,

extended reach drilling (ERD), complex path drilling, casing drilling technology, positive displacement motor technology, and laser technology—which is a newer technology [1–4].

Today's challenge in the field of drilling is to reach deeper targets as fast as possible, minimizing the cost with the continuous improvement of operational efficiency without overlooking health, safety, and environmental guidelines. These factors have rendered drilling activities open to robust techniques and state-of-the-art technologies for an accurate understanding of potential hydrocarbons to be drilled. Improvements in techniques, materials, equipment's, autonomous processes, and advanced technologies have played a role in advancing economically recoverable subsurface resources and reducing the environmental impact at the surface and in the underground. Modern drilling technology is categorized as an advanced suite of both surface tools and downhole tools with the capability of drilling entire well sections in one run with far less impact on the surface and subsurface environments.

The evolution in drilling technologies developed over the last five to ten years for both onshore and offshore development has been focused on several distinct areas of drilling operations, including the monitoring and optimization of surface equipment, drilling methodologies, and downhole tools. Proven technologies seek to reduce both well construction and operational costs while also providing maximum operational efficiency and environmental safety. Some examples of advancements and proven innovation in drilling technology acting as enablers in the discovery of new sources of oil–natural gas and geothermal energy are presented in this article. These existing technologies include horizontal drilling, underbalanced drilling, multilateral drilling, extended reach drilling (ERD), automated drilling, and data analysis. As the topic of onshore and offshore novel developments is extremely broad, this paper sought to provide a synopsis of the Special Issue of MDPI *Energies*, collecting relevant developments that are related to the MDPI-published papers within the last 5 years.

## 2. Enabled Drilling Performance through Advancing Systems and Technologies

The upsurge in the world's energy needs by tapping into hydrocarbons cannot be accommodated without operational drilling systems. As technologies related to drilling systems evolve, the move from the conventional approach to the unconventional technique has enabled the resolution of drilling problems and undetected drilling-related issues through the means of both hardware and software solutions with the integration of business values solutions such as IoT technology, Big Data analytics, Digital Twin, artificial intelligence (AI) and other key technologies transforming energy systems as a way of enhancing operational efficiency during drilling operations and activities [5,6]. The reviews selected below cover industrially proven and academic lab-scale solutions covering multi-functional areas of drilling that are of benefit to well construction and developmental operations as well as to increasing performance to another level.

### 2.1. Company Enabled Drilling Solutions

The trend of automation and other dominant 4.0 technologies such as artificial intelligence, IoT, 3D printing, Big Data, Cloud technology and Digital Twin technology has proven the viability of unlocking business values to improve operations while maintaining customer satisfaction for real-life drilling processes. Many oil and gas companies have leveraged and deployed some of these technologies, thus developing both their own hardware and software tools for operational activity accuracy and decision making for the sustainability of their assets.

**Automation in drilling equipment** such as oil rigs has a strong impact on the exploration and development of hydrocarbons. Many oil companies are now starting to equip their drilling equipment such as oil rigs with autonomous drilling control systems to achieve improved performance times. According to PRNewswire [7], automation helps improve safety standards by reducing human intervention at drill sites and reducing labor costs whilst increasing the operational efficiency of hydrocarbon extraction and being

applicable both offshore and onshore. A typical example is the launching of the first fully autonomous offshore platform capable of performing drilling operations with little or no operational staff by a European oil and gas company [8,9]. Another example is that of integrated drilling systems, such as Amphion and Cyber base, which increase drilling control over the rig equipment, enhance safety and efficiency, and reduce redundancy on rigs by providing control systems that automate drill floor processes and equipment interaction for both offshore and onshore activities [10]. IoT in offshore operations seems to have become an important "added value" concept for most operators [11]. Moreover, as of 2021, automation has been mostly transferred from offshore to onshore activities, particularly those activities with intensive and repetitive activities, such as those involved in shale and coal bed methane drilling. Figure 1 shows that a full automation level is defined as the level at which the computer solely takes care of the whole activity, namely monitoring and data analysis, generating the decisions, selecting the best option, and implementing the selected decision. Although Endsley et al. [12] published this concept about levels of automation two decades ago, Macpherson et al. [13] have adapted Endsley concept for drilling activities, proposing the same 10 levels of automation, whereas the autonomous level involves self-decisional systems such as rotary steerable systems. As of 2021, the new direction is trending towards autonomous drilling rigs that can perform drilling without direct human intervention. The levels of automation shown in Figure 1 are ranked from 1 to 10, where L10 will represent the autonomous drilling system. According to Macpherson et al. [13], the rotary steerable systems are ranked L8–L10, meaning that the need for human intervention is minimal, and only during monitoring/generating of the process. To date, a fully autonomous system, as per the definition provided by Endsley et al. [12], has not yet been realized, however, a few attempts have been already published [14]. Although not comparable, it is worth mentioning that the mining industry has reported fully autonomous blast hole drilling systems since 2018.

Levels of automations as defined in 1999, by Endsley et al.

| | Level of Automation | Monitoring | Generating | Selecting | Implementing |
|---|---|---|---|---|---|
| 1 | Manual Control | Human | Human | Human | Human |
| 2 | Action Support | Human/Computer | Human | Human | Human/Computer |
| 3 | Batch Processing | Human/Computer | Human | Human | Computer |
| 4 | Shard Control | Human/Computer | Human/Computer | Human | Human/Computer |
| 5 | Decision Support | Human/Computer | Human/Computer | Human | Computer |
| 6 | Blended Decision Control | Human/Computer | Human/Computer | Human/Computer | Computer |
| 7 | Rigid System | Human/Computer | Computer | Human | Computer |
| 8 | Automated Decision Making | Human/Computer | Human/Computer | Computer | Computer |
| 9 | Supervisory Control | Human/Computer | Computer | Computer | Computer |
| 10 | Full Automation | Computer | Computer | Computer | Computer |

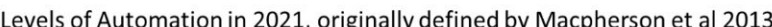

Levels of Automation in 2021, originally defined by Macpherson et al 2013

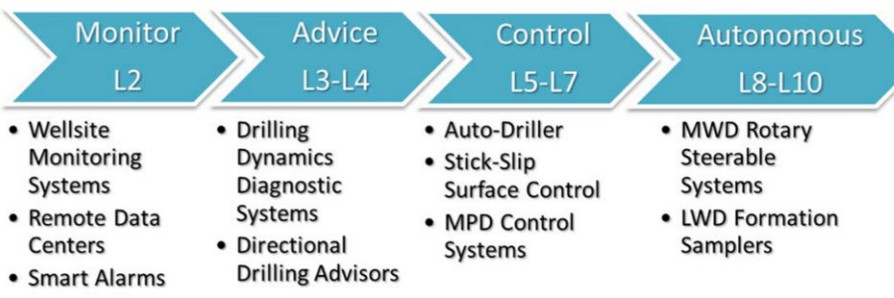

| Monitor | Advice | Control | Autonomous |
|---|---|---|---|
| L2 | L3-L4 | L5-L7 | L8-L10 |
| • Wellsite Monitoring Systems<br>• Remote Data Centers<br>• Smart Alarms | • Drilling Dynamics Diagnostic Systems<br>• Directional Drilling Advisors | • Auto-Driller<br>• Stick-Slip Surface Control<br>• MPD Control Systems | • MWD Rotary Steerable Systems<br>• LWD Formation Samplers |

**Figure 1.** Automation in oil rigs (modified).

In the article by Creegan and Jeffrey [15], an **intelligent drilling optimization application performs as an adaptive autodriller**. This novel technology uses artificial intelligence (AI) algorithms to enhance on-bottom drilling performance. The system features continuous learning capabilities, enabling it to provide proactive drilling dysfunction mitigation

while maximizing the rate of penetration (ROP) and optimizing mechanical specific energy. The system's advantage is that it entails less dependence on human beings in the drilling process, lowering the risk of slow or incorrect responses in drilling dysfunction. During the field application, ROP was improved by 61% and drilling performance—measured as hours on the bottom—was improved by 25%.

Invented over 100 years ago, directional drilling or horizontal directional drilling (HDD) is a trenchless and valuable technology used in oil fields to increase oil production. The rate of technological innovation has never decreased to date, especially not for offshore projects where continuous accumulated experience has served to reduce the total drilling time and the cost of drilling, thus enhancing the profitability of these companies. The trend of automation, propelled by the utilization of 3D visualization tools, has expanded the capabilities of downhole tool technologies, improving the drill bit's performance, and reducing drilling vibrations. One of the new evolutionary alternatives to the traditional rotary steerable system (RSS) is the **Steer-At-Bit Enteq Rotary Tool (SABER)** tool that uses internally directed pressure differentials to steer away from the drill bit face, delivering true "at-bit" geosteering [16,17]. Based on the successful initial testing, the tool does offer a robust, reliable, simple, and cost-effective directional drilling alternative to current RSS options, minimizing downtime whilst maximizing reliability and drilling speeds. For accurate well placement, Haliburton introduced their product called **iCruise, an intelligent push-the-bit RSS**, which allows for improved steerability and drilling performance. This tool is automation-enabled for precise steering and accurate well placement, helping operators reduce well time through faster drilling, reliable performance, and predictable results [18]. To automate decision-making during drilling operations, Motive Drilling Technologies developed a premier **directional drilling bit guidance system**, which automates decision making at the rig. The mentioned merits of this system include drilling time reduction without compromising wellbore quality, which also results in higher productivity. The bit guidance system is built using the latest downhole computer generation, with an improved data-driven automated decision-making algorithm. There are many numbers of this intelligent RSS, for example, the Baker Hughes [19] **i-Trak drilling automation** and others on the market with advanced electronics for tool prognosis, diagnosis and high-speed processors which help directional drillers make efficient drilling decisions and manage vibration in real time.

A better understanding of downhole drilling dynamics and optimization conditions is very vivacious in terms of drilling operations and performance. According to NOV, an improper BHA design leads to lower drilling efficiency, which implies the need for BHA redesign at high costs and time lost. To reach the borehole total depth (TD) with excessive drag and reduced tortuosity at the end of the well, NOV developed the **SelectShift downhole adjustable motor** that allows higher surface RPM, improved hole condition and cleaning, minimizing spiraling/tortuosity and improving ROP [20]. Another innovative BHA optimization tool is the Schlumberger [21] **OptiDrill real-time drilling intelligence tool,** which is capable of collecting a wide range of surface and downhole data and using advanced algorithms and enables event detection and customized reporting. This system was developed to allow operators to mitigate risk, reduce premature tool failure, and upsurge downhole drilling efficiency.

Well control is another key and crucial sub-sectional area of the drilling operation. The blowout incident from the Macondo oil field and past blowout accidents have reflected the need for rigorous safety measures, ensuring safe and environmentally responsible drilling operations. Devising some well control automation systems will enable fast identification, decision making, and reaction to well control events. Some of the latest technology aiming to reduce well control risks led to the development of **automated well control solutions** by Safe Influx. It has been successfully demonstrated that this system can provide support to the driller, dramatically reducing our exposure to risks from human factors. The system has the ability to detect the presence of a fluid influx condition in a wellbore, make a decision against criteria to favor a shut-in, and then automatically initiate an initial well

control protocol that results in the well-being safely shut-in. This revolutionary technology is capable of reducing the size of an influx compared to conventional techniques. This implies reductions in delays, costs, and operational issues in getting back to drilling [22,23]. The combination of these technologies will provide an automated secondary well control which will result in enhanced well control and integrity while improving the drilling efficiency [24].

### 2.2. Academic Research-Enabled Drilling Solutions

Academic research and development have made significant contributions towards enhancing drilling operations. These validated and proven technical solutions have had a tremendous effect on improving drilling efficiency and cutting operational costs through radical innovation and digital transformation in collaboration with industries and private institutions. Highlighted below are some of the selected solutions that transcend subject areas such as drill bit technology, drilling optimization, instrumentation and automation, well design and smart well, well cementing, and the application of Big Data and data analytics to the benefit of drilling operations.

Sharma et al. [25] designed and developed an **instrumentational laboratory-scale test rig (stick-slip simulator)** to identify and predict downhole vibration events such as torsional vibrations. According to Sharma et al. [25], "the setup is capable of safely recreating drilling vibrations that occur in wells, including stick-slip vibrations, which are detrimental in nature". Their experimental test rig was built to analyze drill string vibrations using the so-called mechatronic concept, which is a combination of mechanical and electrical components fully controlled by complex real-time software. Their results highlighted that the measured vibrations modes are a function of various parameters such as rotational speed (RPM), torque, and weight on bit, as well as the bit sticking time period and frequency. In their study findings, sampling rates below 10 Hz were found to be unsuitable for the correct identification of the severity of stick-slip vibrations. Furthermore, the study shows the necessity of the good integration of hardware and software to achieve reliable results. The quality of the sensors and their sampling rate were found the most crucial factors in the design of the experimental setup.

Braga et al. [26] presented a conceptual methodology and system to **predict bit in real time with 30-second** updates using WITMSL data, standard MWD directional surveys sent once per stand, and specific BHA data. The system also provides context, plotting the projections in relation to the well plan, drilling window, and formation tops. In their conclusion, the algorithm's performance, when used to project with steerable motor or RSS assemblies, has a median divergence of less than two ft. for all subject wells. The Q3 divergence values were less than three ft. for steerable motor projections and less than four ft. for RSS. The flexibility of the proposed method was validated in four horizontal wells in West Texas, USA, using 839 survey stations to validate the results. The resulting median divergence considering all validated projections was less than a foot.

Koc and Taleghani [27] presented a new method to **estimate the critical depth of cut (DOC) of different rock specimens** (from ductile to brittle failure and vice versa) which consists of "measuring the roughness of the groove for determining the transition point of failure modes for every rock sample after the scratch test". The method was accomplished through extensive laboratory tests in parallel with numerical solutions. According to the authors, "the average change in the surface roughness (Rt) versus the scratched surface roughness (ΔR) can be used to identify the rock failure mode and determine the transition point for the cutting process. The value of this slope increases until the depth of cut reaches the transition point, and then the slope reaches a constant value". Another important achievement of their proposed model is that, for the first time, the DOC is proposed and measured. The tests show that the measured surface roughness acts identically to the tangential measured forces.

Nystad et al. [28] investigated the **application of a data-driven optimization method called extremum seeking (ES)** to attain an improved and safer drilling process using a

novel automatic real-time approach based on the minimization of the mechanical specific energy (MSE). The ES algorithm collects a wide range of information to assess the current downhole conditions. Additional parameters are generated during drilling changes in the applied weight on bit (WOB) and the drill string rotational rate (RPM). The process is performed automatically, and the process is optimized using downhole real-time information. The proposed algorithm is said to handle various drilling constraints related to drilling dysfunctions and hardware limitations especially when ROP and torque parameters are limited. The methodology is yet to be field validated, but the authors are expecting an ROP improvement of up to 20–170%.

Thakur and Samuel [29] established a novel method to **predict downhole data by employing deep learning using surface data** that can improve wellbore placement and increase drilling efficiency by improving the rate of penetration (ROP) and reducing downtime caused by tool failure. In their approach and for initial prediction, the model was trained on analog wells. In the absence of an analog well, surface WOB data can also be used as estimates of downhole WOB. The next step of their method was the use of relatively inexpensive technology to collect downhole data, on mud pumps for example, and sync the data in time. Then, a new model is trained, or the trained model is updated for the existing well. The model is then used to predict downhole data for the current well. The model can be updated as more and more data become available for better prediction. The novel study is yet to be validated on the field; however, the developed model has a median error as low as 3% and can accurately predict downhole data in real-time with the prediction accuracy varying from well to well and drilling depths. The model is also very robust to the amount of noise or outliers present in the data and can predict downhole conditions 50–60 ft. ahead with reasonable accuracy. The results demonstrate how deep learning can be cost-effectively employed for downhole data prediction.

Dumitrescu et al. [30] conducted a numerical performance analysis for the **corrosion repair of pipelines using a modern composite materials system**. The authors revealed that composite repair systems are highly effective in restoring corroded pipes when the selected repair materials have consistent mechanical properties that are compatible with the base steel pipe. For example, a composite material with a Young modulus equal or similar with steel pipe seems to yield the best results. According to their finite elements simulation results, the width of the corroded area has limited influence on the stress state of the repaired zone. They further emphasized the following observations from the finite element analysis of the influence of the defect orientation and fillet radius upon the stress distribution: machining the defect area as an inclined rectangle will reduce the preparation time without influencing pipe safety and the angle of the defect orientation will increase the von Misses stresses by only 10%.

Sliwa et al. [31] conducted a study to improve the impact understanding of rotational speed, drill bit diameter, and air pressure on drilling velocity for the down-the-hole (DTH) drilling method, for a given lithology. The study approach was validated using well information from the Lubin Basin in Poland with such an enhanced solution aiding drilling efficiency and process for wellbore heat exchanger (BHE) installations for selected locations. To conduct the study, nine boreholes with common geological profiles were drilled using the rotary down-the-hole drilling with the air method. A down-the-hole hammer (DTH) with a diameter of 4 inches was used and the horizontal distance between the boreholes was about six meters. The following conclusions were drawn from their study: "for every analyzed situation, the highest drilling velocity is achieved when the greatest drilling pressure is applied." The authors pointed out that the lowest ROP was recorded at the lowest drilling pressure. Additionally, they noticed that the increase in air pressure in the drill pipe causes an increase in the consumption of feed energy, resulting in high operating costs. The test also demonstrates that the DTH ROP also depends on the rotational speed of the drill bit. The lithology was found to have a major impact on the ROP.

The study by Fang et al. [32] addressing the narrow density window achieves effective control of the downhole pressure in complex geological environments, leading to the **de-**

**velopment of an MPD wellbore transient flow dynamic model** using the "compressible gas–fluid two-phase flow with the drift flux" model. The developed model was built to accurately describe the characteristics of transient multiphase flow in the wellbore. To verify the established model, a simple experimental device was designed to accommodate a wellbore simulator, liquid circulation system, and an air supply unit. In their findings, the pressure followed a trend of a gradual decline, sharply followed by a pressure fall induced by the original fluid being dispelled with the ingress of gas before an incremental increase again. The control of wellhead backpressure is performed using the throttle valve. Higher wellhead backpressure means lower downhole pressure. The authors have stated that "When the gas–liquid two-phase flow in the wellbore reaches an equilibrium state, the downhole pressure will decrease less with the increase of drilling fluid displacement, and the time of gas reaching the wellhead will be earlier".

Table 1 highlights the technologies covered in this paper.

**Table 1.** A summary of the technologies covered in this paper is shown herein.

| Author | Technology | Comments |
|---|---|---|
| Endsley et al. (1999) | Definition of autonomous drilling | This is the first accepted version of a definition for autonomous drilling |
| Macpherson et al. (2013) | Upgrading the autonomous drilling definition and steps forward | This is the most accepted approach by the industry toward automated drilling systems |
| Cayeux et al. (2021) | Proposes an autonomous drilling algorithm | A complex autonomous drilling algorithm is presented |
| Creegan and Jeffrey (2020) | Intelligent drilling optimization application performs as an adaptive autodriller | This is the introduction of the autodriller using recent optimization techniques |
| oedigital article (2019), Enteq, Baker, Schlumberger, | Various rotary steerable systems are introduced with an autonomous mode | The current market is dominated by the low-cost high-performance RSS with autonomous options |
| Sharma et al. (2020) | Instrumentational laboratory-scale test rig (stick-slip simulator) | A mechatronic integration in a unique experimental setup to understand and mitigate stick-slip situations at the bit |
| Braga et al. (2021) | Predicts bit in real time with 30 s updates using WITMSL data | A very useful integration of real-time data at rig site with bit predictive software package |
| Koc and Taleghani (2020) | Estimates the critical depth of cut (DOC) of different rock specimens | The proposed method can be integrated in ROP evaluation software package and will help in understanding when bit wear is high |
| Nystad et al. (2021) | Application of a data-driven optimization method called extremum seeking (ES) | Best applicable to improve safety of drilling process |
| Thakur and Samuel (2021) | Predict downhole data by employing deep learning using surface data | Focused on improving rate of penetration |
| Dumitrescu et al. (2021) | Corrosion repair of pipelines using a modern composite materials system | Proposed experimental and numerical work to estimate the repair quality of composite repair systems |
| Sliwa et al. (2020) | Understanding of the impact of rotational speed, drill bit diameter, and air pressure on drilling velocity for the down-the-hole (DTH) drilling method | The method is useful to better define automated downhole hammer drilling processes |
| Fang et al. (2019) | Development of an MPD wellbore transient flow dynamic model | Better prediction of MPD and helping in automated MPD control |

*2.3. Other Next-Generation Ideas That Affect Drilling*

Wellbore construction is not only drilling the hole but the ability to create a safe environment for the wellbore completion for the life of the well. The well construction key element is the casing-cement system that will consolidate the well and isolate unwanted strata. With novel advancements in the production and reservoir technologies as well as in renewable such as geothermal energies, the life of the well is pushed to exceed 25 years. With this in mind, long-term casing-cement properties are now the new research focus. Several researchers have pointed out the importance of long-term cement properties [33–37].

Kremieniewski [35] investigated the significant influence of graphene oxide (GO) on the rheological properties of selected cement slurries. The conducted studies are important for the oil and gas industry through the better optimization of the cementing process. The author's findings show "that the graphene oxide admixture to the cement slurry resulted in an increase of its rheological parameters". The addition of graphene oxide led to very small increase in slurry viscosity. The lowest measured values of rheological parameters, for all tested slurries, were noticed for the cement slurries with 0% graphene oxide admixture. The author concluded that the "concentration range of 0.01 to 0.03% GO was optimal for cement slurry technology because it caused an improvement for the slurry's mechanical parameters parallel to maintaining the rheology on the required level." The experimental work of Kremieniewski [35] was also supported by results on cement properties by others, including Konsta et al. [38], Ranjbortareh et al. [39] and Kasiravalad [40].

Eltsov and Patzek [36] proposed a technique for the detection of the integrity of magnetic cement behind casing made of composite fiberglass. The study objective was to demonstrate that a specially designed magnetic logging tool is capable of detecting small changes in the magnetic permeability of cement through a non-magnetic tubular. The proposed tool and methodology will enable identifying potential cement debonding zones. It was revealed in their conclusions that the optimum tool length was between 0.25 and 0.6 m. It was also mentioned that the frequency range between 0.1 and 10 kHz will offer the most accurate results. According to Eltsov and Patzek [36] "Signal phase at a high frequency was more sensitive to cement solidification, rather than amplitude. Cavities and cracks filled with magnetic cement were visible on the logs." Using radial distributed sensors, additional cement defects could be revealed. However, the method is applicable to non-magnetic tubulars only, and it will enable evaluating the cement quality in moderate temperature wells (<150 °C), as well as determine the poor zonal isolation and cement hardening.

Arbad et al. [41] have shown the importance of measuring the bonding between casing and cement and compared these properties with other cement mechanical features such as shear strength and unconfined compressive strength. The paper highlights first a simple yet effective method to measure and compare the so-called interfacial bonding shear strength of casing–cement interaction as well as pure shear strength. The main outcome of the paper was that the bonding strength did not increase after 7 days of curing at all, resulting in a limited stress available to hold casing in place.

Arbad et al. [42] presented the study of possible contamination effects on oilwell cements, highlighting that at low concentrations and a short curing time, there is no evidence of any influence, but long-term effects are critical, with massive cement properties reduction. The main conclusion of this paper is that the next generation of well constructions needs to focus on better and cleaner wellbore fluid solutions with minimum cement contaminations, which will result in better long-term integrity for the life of the well.

All the above works on cement show the importance of the future study of novel and more reliable materials that could improve well integrity and lower overall well construction costs. These solutions are highly important and can achieve maximum applicability through drilling automation and careful process management.

### 3. Unconventional Solutions: Bridging the Gaps and Catalyst for Change

The evolution of drilling technologies has been a long process. Since drilling operations remain the most critical, complex, and costly operation during well development, decreasing the gap between conventional technology and emerging technologies through automation will enhance and improve drilling activities by improving drilling efficiency and increasing safety for all personnel without making costly errors. Despite the deployment and tapping into some 4.0 technologies, the digitalization of drilling operation and management systems is still in the early stages when compared to other industries such as aviation, automobile, and transportation. Technologies such as mixed reality, robotics and drones, Digital Twin and unmanned aerial vehicles are yet to be fully adopted, while they are already quickly growing in terms of application and gaining trust in other industries. Utilizing and leveraging these emerging technologies within an autonomous drilling operation will improve consistency, reduce operating costs, and lower risk potential. Additionally, the new frontier of change from conventional drilling technology will be able to adapt to complex drilling operations which can lead to costly nonproductive time.

As the oil and gas industry is fully entering the world of automation and digital process control, it has been predicted to have more digital instrumentation and automated process control as compared to the traditional drilling method. A solution approach across robotic technology will be a game-changer to drilling operation, especially the drill floor. Having a smaller human footprint with more digital instrumentation will help comply with health, safety, and environmental issues, and would be more cost-effective. Robotic drilling systems (RDSs) are designed to allow unmanned drilling operations. In addition, state-of-the-art robotic drilling technology will be capable of automatic rig move operations. Another emerging solution is that of unmanned aerial vehicles (UAVs). UAVs could be an alternative for the inspection of the mud and storage tanks, transport pipelines, and complex equipment used in refineries of the oil–gas–geothermal facilities. Drones are generally operated using ground control centers, but their operation needs robust flight control techniques combined with state-of-the-art inertial navigation, data processing, and tracking control. The addition of the latest technology will make the future drones more effective for the exploration of oil and gas reservoirs, especially those located in environments that are unsuitable for human access [43].

Figure 2 shows the current adaptation technologies as related to drilling operations and management. Currently, we are in the semi-matured technology phase, and it has been proven to be advantageous in drilling operations. Pushing beyond traditional limitations through technological innovation and leveraging on these much-required technologies will improve the operational consistency in the future, reduce nonproductive time and cost overruns across drilling activities. This reflects a commitment to technical authority, engineering excellence, and safety. Additionally, utilizing these technologies will play a critical role in meeting energy global demand and enable the discovery of new resources, regardless of the operational condition that was not previously economical to produce.

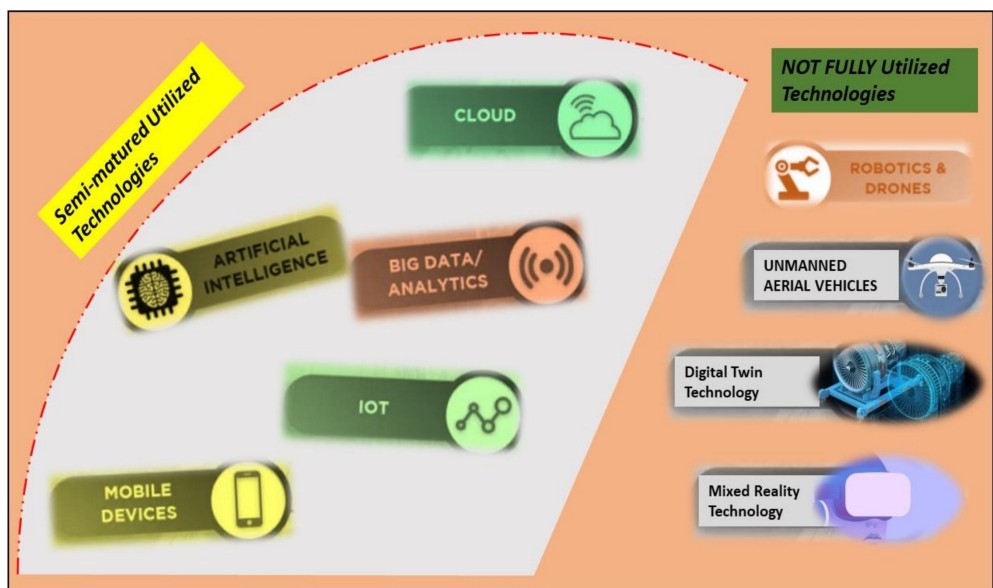

**Figure 2.** Emerging technologies adoption in drilling operations.

## 4. Conclusions

Drilling is a highly dynamic process and has multiple needs that can be addressed through the increased adoption of automation and control due to the complexity of drilling operations.

In this article, we expounded the necessity of leveraging emerging technologies that will change the trajectory in the field of drilling to reduce both well construction and operational costs, while also providing maximum operational efficiency and environmental safety. Additionally, selected companies and academic research that have enabled drilling performance through advancing technologies were identified.

Results have shown that utilizing this novel technology allows faster decision making, tracking new business opportunities, reorganizing operations, and significantly reduce well control risks and operational costs. With a strong partnership between oil–gas–geothermal industries and academia, there are high probabilities of developing more novel and profound tools and technologies to address unsolvable drilling problems, thus increasing discoveries of oil–gas–geothermal sources in highly challenging locations.

**Author Contributions:** Conceptualization, C.T. and O.B.; methodology, C.T. and O.B.; validation, C.T. and O.B.; formal analysis, C.T. and O.B.; investigation, C.T. and O.B.; data curation, O.B.; writing—original draft preparation, C.T. and O.B.; writing—review and editing, O.B. and C.T.; supervision, C.T.; project administration, C.T. and O.B. Both authors have read and agreed to the published version of the manuscript.

**Funding:** This research received no external funding.

**Institutional Review Board Statement:** Not applicable.

**Informed Consent Statement:** Not applicable.

**Conflicts of Interest:** The authors declare no conflict of interest.

## Nomenclature

| | |
|---|---|
| BHA | Bottomhole Assembly |
| MPD | Managed Pressure Drilling |
| MWD | Measurement While Drilling |
| WITSML | Well-Site Information Transfer Standard Markup Language |
| RSS | Rotary Steerable System |

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
