# Peer review of "An Outlook of Drilling Technologies and Innovations: Present Status and Future Trends"

_energies, doi:10.3390/en14154499_

Round 1

Reviewer 1 Report

A summary paper presenting briefly existing results from industry and research related to improvements on drilling technologies such as: directional drilling, adaptive auto-driller, well control, monitoring, lab test rigs, predictive methods for bit behaviour, corrosion. The paper addresses open problems in the field and possible ways forward in Section 3.

Line 104-105: it would improve the text to provide a reference to the autonomous offshore platform example mentioned here, if possible.

As a general comment, it would improve the text to define in the beginning of the paper what is meant by autonomous drilling throughout the paper: is it meant robotics/mechanization at the level of the surface equipment or an actual autonomous decision-making system that is taking decisions by itself, also considering real-time optimization of  the down-hole conditions. (Reference levels of automation: SPE-166263-PA).

Line 152-154: It would improve the text to provide references if possible.

Section 2.2. Academic research enabled drilling solutions: A nice short summary of a limited number of academic results, most of them on simulation of the physical processes in drilling and which were tested in the lab. I encourage the authors to investigate further the published results on automated drilling and autonomous drilling (here autonomous as in SPE-166263-PA), also for technologies which have been validated in the field (this is a general comment and it should not influence current paper acceptance/revision).

Line 352: It should be 3.2 instead 2.2.

Author Response

Reviewer 1.

A summary paper presenting briefly existing results from industry and research related to improvements on drilling technologies such as: directional drilling, adaptive auto-driller, well control, monitoring, lab test rigs, predictive methods for bit behaviour, corrosion. The paper addresses open problems in the field and possible ways forward in Section 3.

Line 104-105: it would improve the text to provide a reference to the autonomous offshore platform example mentioned here, if possible.

Answer: Thank you for your comment. we have added 2 references about this concept.

As a general comment, it would improve the text to define in the beginning of the paper what is meant by autonomous drilling throughout the paper: is it meant robotics/mechanization at the level of the surface equipment or an actual autonomous decision-making system that is taking decisions by itself, also considering real-time optimization of  the down-hole conditions. (Reference levels of automation: SPE-166263-PA).

Answer: Thank you for your comments. We have changed the figure 1 and included a short paragraph about autonomous concepts. We highly value your recommendation.

Line 152-154: It would improve the text to provide references if possible.

Answer: Thank you for your comments, references have been included.

Section 2.2. Academic research enabled drilling solutions: A nice short summary of a limited number of academic results, most of them on simulation of the physical processes in drilling and which were tested in the lab. I encourage the authors to investigate further the published results on automated drilling and autonomous drilling (here autonomous as in SPE-166263-PA), also for technologies which have been validated in the field (this is a general comment and it should not influence current paper acceptance/revision).

Answer: Thank you for your comment. We have tried to stay within 5 years time frame and our focus was highly on papers published within MDPI or covering topics that has been published, since this paper should conclude our special issue called Drilling technologies for future generations. We, however, fully understand your comment, and we tried to add some more refs.

Line 352: It should be 3.2 instead 2.2.

Answer: thank you for your comment, we have corrected our mistake.

Reviewer 2 Report

Authors attempt to provide their insight regarding the current and future technological developments and trends in drilling industry and academia. The topics that the authors have pointed out are valid and sound. However, drilling is extremely large and broad topic in which many other topics and branches are not mentioned, and this is totally understandable and acceptable. Some topics including advances in drilling fluids, data transfer, wellbore stability, shale instability, cementing, trajectory control, etc can be also mentioned. But, the topics that the authors have selected are also important.

One major critic that can be pointed out is; authors have referred to their own work too many times. They have cited their past work substantially, which makes the references look a little "cheap". Also, number of references for such a review paper is expected to be much more than 32.  

Authors are encouraged to increase the number of references, and decrease their own referrals.

Author Response

Reviewer 2:

Authors attempt to provide their insight regarding the current and future technological developments and trends in drilling industry and academia. The topics that the authors have pointed out are valid and sound. However, drilling is extremely large and broad topic in which many other topics and branches are not mentioned, and this is totally understandable and acceptable. Some topics including advances in drilling fluids, data transfer, wellbore stability, shale instability, cementing, trajectory control, etc can be also mentioned. But, the topics that the authors have selected are also important.

One major critic that can be pointed out is; authors have referred to their own work too many times. They have cited their past work substantially, which makes the references look a little "cheap". Also, number of references for such a review paper is expected to be much more than 32.  

Authors are encouraged to increase the number of references, and decrease their own referrals.

Answer: Thank you for your comment. We understand your need for much more references, however, after we initiated a linkedin pool, we have found out that most of the answers were showing that even a review paper does not need a high number of citations, especially that most of the readers will not focus on the references unless the paper is strongly showing their importance. Furthermore, our paper is concluding the special issue of MDPI Energies, and thus our focus is on the past 5 years most important technologies and how these technologies are related with papers published within the SI and MDPI papers as well. However, we have increased the number of total references to over 36 while deleting some of the self citations as per your recommendation.

Reviewer 3 Report

  1. It is better to strength the objectives of this paper in particular in the last paragraph in Introduction. In lines 68 to 77, readers will not be clear whether the authors try to summarize onshore and offshore development (this is a broad topic) or just present some new technologies. Point by point is suggested here so audience could easily catch the objective of this paper. In addition, the title of this paper include two keywords, present status and future trends but it is unclear to see the authors how to organize these key points in Introduction.
  2. In line 101, is it Figure 1 or 2?
  3. The authors did not clearly address Figure 1. What is meaning of High and Medium? What is 300+ industries for major trends and challenges and so on?
  4. In lines 119 and 120, it is ambiguous and those improvements come from which field application?
  5. Although the authors listed those new emerging technologies and equipment and categorized them, it is still difficult to catch them. The suggestion include: (1) Rather using company enabled drilling solutions and academic research enabled drilling solutions, the authors could divide those development into several aspects, for example, drilling bits, steering, data logging, inspection and well completion (the authors could adjust contents corresponding to specific areas) and listed those new technologies in those aspects so readers could easily follow and find those contents which they are interested in.

Others:

  1. There is a period missing in line 77.
  2. The authors should thoroughly check the references. For example, where is Dumitrescu et al. (2002), Thakur and Samuel (2021)?

Author Response

Reviewer 3:

  1. It is better to strength the objectives of this paper in particular in the last paragraph in Introduction. In lines 68 to 77, readers will not be clear whether the authors try to summarize onshore and offshore development (this is a broad topic) or just present some new technologies. Point by point is suggested here so audience could easily catch the objective of this paper. In addition, the title of this paper include two keywords, present status and future trends but it is unclear to see the authors how to organize these key points in Introduction.

Answer: Thank you for your comment. We have added a new paragraph at the end of introduction with the hope to fully answer to your question.

  1. In line 101, is it Figure 1 or 2?

Answer: Thank you for your comment, it is indeed Figure 1.

  1. The authors did not clearly address Figure 1. What is meaning of High and Medium? What is 300+ industries for major trends and challenges and so on?

Answer: we have decided to remove the figure 1 and replace it with a new figure that better fits our paper. The new figure 1 was fully described in a new paragraph.

  1. In lines 119 and 120, it is ambiguous and those improvements come from which field application?

Answer: we have re-orderd our text to highlight the field application

  1. Although the authors listed those new emerging technologies and equipment and categorized them, it is still difficult to catch them. The suggestion include: (1) Rather using company enabled drilling solutions and academic research enabled drilling solutions, the authors could divide those development into several aspects, for example, drilling bits, steering, data logging, inspection and well completion (the authors could adjust contents corresponding to specific areas) and listed those new technologies in those aspects so readers could easily follow and find those contents which they are interested in.

Answer: Thank you for your comments and we fully understand your point of view. However, our intent and decision is to highlight in two separate tracks the developments made by the industry and the help offered by academic institutions. We search for a similar approach, and we believe that we are the first proposing such a review paper in a such structure. We hope that our structure will be accepted by the reviewer.

Others:

  1. There is a period missing in line 77.

Answer: Thank you for your comments. We have corrected this.

  1. The authors should thoroughly check the references. For example, where is Dumitrescu et al. (2002), Thakur and Samuel (2021)?

Answer: Thank you, we have included these references.

Round 2

Reviewer 2 Report

Authors have addressed the previous comments and improved the referencing as well as pointed out the "selected topics" about automation.

Author Response

thank you. we are happy to be able to answer all your comments.

Reviewer 3 Report

  1. Although the authors have changed into a new figure 1, this figure is still not clear. For example, what do 'L' and those numbers stand for? No information is provided in the context or figure.
  2. One or two tables should be provided for Company enabled drilling solutions and academic research enabled drilling solutions. In the tables, summaries from those references and what catogories those technologies belong to should be included. Readers could easily track the interesting part for them easily at least.  

Author Response

Reviewer 3.

  1. Although the authors have changed into a new figure 1, this figure is still not clear. For example, what do 'L' and those numbers stand for? No information is provided in the context or figure.

Thank you for your comments, we have add a paragraph explaining the figure as per your request. L stands for Readiness Level of automation.

  1. One or two tables should be provided for Company enabled drilling solutions and academic research enabled drilling solutions. In the tables, summaries from those references and what catogories those technologies belong to should be included. Readers could easily track the interesting part for them easily at least.  

Thank you for your suggestion. Although we believe that such a table is not necessary, we have introduced a final table summarizing our technologies.

Author

Technology

Comments

Endsley et al (1999)

Definition of autonomous drilling

This is the first accepted version of autonomous drilling definition.

Macpherson et al (2013)

Upgrading the autonomous drilling definition and steps forward

This is the most accepted approach by the industry toward automated drilling systems

Cayeux et al 2021

Proposes an autonomous drilling algorithm

A complex autonomous drilling algorithm is presented

Creegan and Jeffrey (2020)

intelligent drilling optimization application performs as an adaptive autodriller

This is the introduction of the autodriller using recent optimization techniques.

Pallanich 2019, Enteq, Baker, Schlumberger,

Various Rotary steerable systems are introduced with autonomous mode

The current market is dominated by the low cost high performance RSS with autonomous options

Sharma et al (2020)

instrumentational laboratory-scale test rig (stick-slip simulator)

A mechatronic integration in a unique experimental setup to understand and mitigate stick slip situations at the bit.

Braga et al (2021)

predict bit in real-time with 30-second updates using WITMSL data

A very useful integration of real time data at rig site with bit predictive software package.

Koc and Taleghani (2020)

estimate the critical depth of cut (DOC) of different rock specimens

The proposed method can be integrated in ROP evaluation software packaged and will help in understanding when bit wear is high.

Nystad et al (2021)

application of a data-driven optimization method called extremum seeking (ES)

Best applicable to improve safety of drilling process

Thakur and Samuel (2021)

predict downhole data by employing deep learning using surface data

Focused on improving rate of penetration

Dumitrescu et al (2021)

corrosion repair of pipelines using a modern composite materials system

Proposed experimental and numerical work to estimate the repair quality of a composite repair systems

Sliwa et al (2020)

understanding of the impact of rotational speed, drill bit diameter and air pressure on drilling velocity for the down the hole (DTH) drilling method

The method is useful to better define automated downhole hammer drilling processes.

Fang et al (2019)

development of an MPD wellbore transient flow dynamic model

Better prediction of MPD and helping in automated MPD control